

# TextCenGen: Attention-Guided Text-Centric Background Adaptation for Text-to-Image Generation

**Tianyi Liang** [†123]  **Jiangqi Liu** [†1]  **Yifei Huang** [1]  **Shiqi Jiang** [1]  **Jianshen Shi** [1]  **Changbo Wang** [1]  **Chenhui Li** [*1]

## Abstract

Text-to-image (T2I) generation has made remarkable progress in producing high-quality images, but a fundamental challenge remains: creating backgrounds that naturally accommodate text placement without compromising image quality. This capability is non-trivial for real-world applications like graphic design, where clear visual hierarchy between content and text is essential. Prior work has primarily focused on arranging layouts within existing static images, leaving unexplored the potential of T2I models for generating text-friendly backgrounds. We present TextCenGen, a training-free dynamic background adaptation in the blank region for text-friendly image generation. Instead of directly reducing attention in text areas, which degrades image quality, we relocate conflicting objects before background optimization. Our method analyzes cross-attention maps to identify conflicting objects overlapping with text regions and uses a force-directed graph approach to guide their relocation, followed by attention excluding constraints to ensure smooth backgrounds. Our method is plug-and-play, requiring no additional training while well balancing both semantic fidelity and visual quality. Evaluated on our proposed text-friendly T2I benchmark of 27,000 images across four seed datasets, TextCenGen outperforms existing methods by achieving 23% lower saliency overlap in text regions while maintaining 98% of the semantic fidelity measured by CLIP score and our proposed Visual-Textual Concordance Metric (VTCM).

## 1. Introduction

In graphic design, achieving a harmonious visual effect between text and imagery is essential for clear expression. The choice of background can influence text visibility and comprehension. A common design objective is to place text within an image in a way that is both visually pleasing and clearly conveys the intended message. A preferred strategy is to position the text in the golden ratio, which is believed to be aesthetically optimal. However, designers often grapple with the issue of backgrounds that compete with or obscure the text, as illustrated in Figure 1😵, where unsuitable backgrounds detract from the text's readability and aesthetic appeal, regardless of any adjustments to text color or size. We aim to facilitate the creation of text-friendly images (see Figure 1😋), ideal for text placement and meeting the growing demand due to the increasing use of Text-to-Image (T2I) models for background graphics.

Traditional approaches to graphic design, especially in poster creation, have largely focused on arranging layouts in static natural background images, elements, and text (Guo et al., 2021; Cao et al., 2012; O'Donovan et al., 2014; Li et al., 2022). However, producing text-friendly images remains a challenge due to the complexity of background elements. Our insight, derived from Figure 1, reveals that a clear separation between the main objects and the text areas is essential. Recent advancements in diffusion-related research have shown that it is possible to manipulate the primary objects within cross-attention maps, making the adaptation of background images to accommodate text a feasible endeavor (Hertz et al., 2022; Epstein et al., 2023; Wang et al., 2024). However, we discovered that directly reducing attention in the target region leads to a semantic reduction in the generated image's match with the prompt. *Could we move the conflicting objects out of the target area before reducing attention?*

In response, we introduce **TextCenGen**📝[1], as illustrated in Figure 1, a new method that employs cross-attention maps and force-directed graphs for effective object placement and whitespace optimization. We also implement

---

[†]Equal contribution  [1]College of Computer Science and Technology, East China Normal University, Shanghai, China [2]Shanghai Institute of AI Education, Shanghai, China [3]Shanghai Artificial Intelligence Laboratory, Shanghai, China. Correspondence to: Chenhui Li <chli@cs.ecnu.edu.cn>.

*Proceedings of the 42nd International Conference on Machine Learning*, Vancouver, Canada. PMLR 267, 2025. Copyright 2025 by the author(s).

---

[1]Open source code at: https://github.com/tianyilt/TextCenGen_Background_Adapt

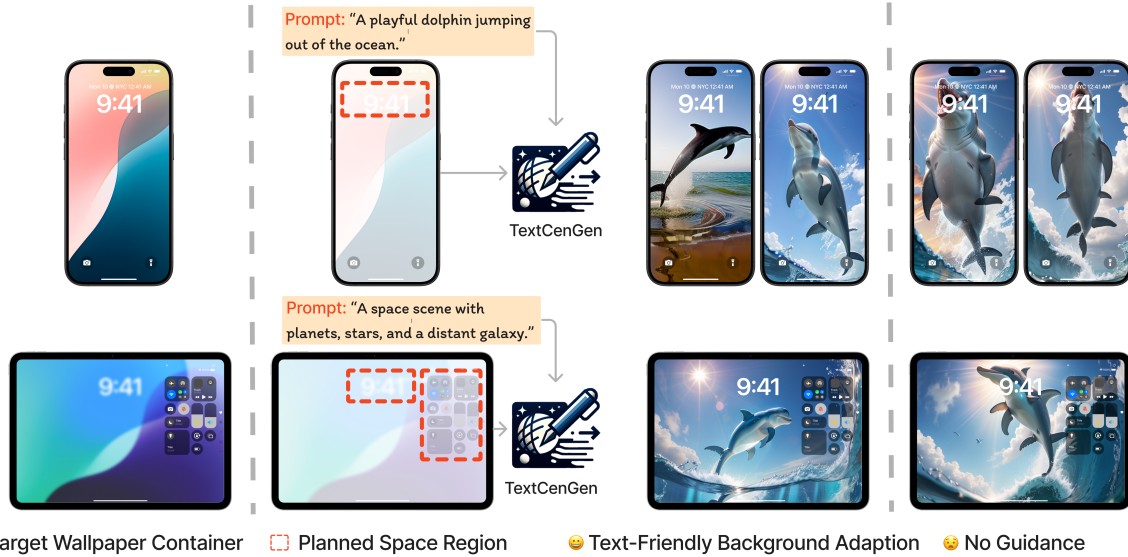

Target Wallpaper Container    [ ] Planned Space Region    😊 Text-Friendly Background Adaption    😕 No Guidance

*Figure 1.* TextCenGen is a training-free method designed to generate text-friendly images. By using a simple text prompt and a planned blank region as inputs, TextCenGen creates images that satisfy the prompt and provide sufficient blank space in the target region. For example, the text-friendly T2I approach helps users customize their favored text-friendly wallpapers for mobile devices with any T2I model, avoiding visual confusion caused by the main objects overlapping with UI components.

a spatial excluding cross-attention constraint to ensure smooth attention in areas designated for text. To establish a new benchmark for this innovative task, we constructed a diverse dataset gathered from three unique sources, along with five evaluation metrics, to comprehensively assess the performance. The contributions of our paper are three-fold:

- We propose a new task of text-friendly T2I generation, which creates images that satisfy both the prompt and reserve space for pre-defined text placements. The task consists of a benchmark including a specialized dataset and tailored evaluation metrics.

- We introduce **TextCenGen**, a plug-and-play, training-free background adaptation framework for dynamic text placement in generated images.

- We develop force-directed cross-attention guidance, adaptable to various attention mechanisms across different T2I models, ensuring a harmonious layout of text and imagery.

## 2. Related Work

**Text Layout of Natural Images** has evolved significantly, transitioning from traditional layout designs to more advanced methods influenced by deep learning. Initially, poster design focused on creating layouts with given background images, elements, and text (Guo et al., 2021; Cao et al., 2012; O'Donovan et al., 2014; Li et al., 2022; Zhang

et al., 2020). The integration of deep learning in text layout of natural image generation has led to various models such as GAN (Goodfellow et al., 2014; Zheng et al., 2019; Li et al., 2019; Zhou et al., 2022), VAE (Jyothi et al., 2019), transformers (Vaswani et al., 2017; Inoue et al., 2023a; Wang et al., 2023) and diffusion models (Ho et al., 2020; Hui et al., 2023; Inoue et al., 2023b; Chai et al., 2023; Li et al., 2023a). These models have been instrumental in learning layout patterns from large datasets. Subsequent research explored methods to retrieve matching background images based on text and image description (Jin et al., 2022). Since the development of image editing method based on diffusion model, scene text generation methods such as TextDiffuser (Chen et al., 2023a;b) and DiffText (Zhang et al., 2024) have addressed the challenges of generating clear text with diffusion models. However, these methods often rely on the presence of a "sign" or similar element within the prompt (e.g., a T-shirt) to place text. They do not explicitly tackle the problem of creating text-friendly images where the background itself is crafted to adapt pre-defined text regions. Our approach extends these capabilities by allowing the primary objects in generated images to yield space to text regions, resulting in more harmonious and aesthetically pleasing compositions.

**Text-to-Image Generation** has advanced with diffusion models (Ho et al., 2020), producing realistic images and videos that align closely with text prompts (Rombach et al., 2022; Singer et al., 2023; Chen et al., 2023c; Esser et al., 2024). Innovations in this field include GLIDE (Nichol

et al., 2022), which integrates text conditions into the diffusion process, Dall-E 2 (Ramesh et al., 2022) with its diffusion prior module for high-resolution images, and Imagen (Saharia et al., 2022), which uses a large T5 language model to enhance semantic representation. Stable diffusion (Rombach et al., 2022) projects images into latent space for diffusion processing. Beyond text conditions, the manipulation of diffusion models through image-level conditions has been explored. Methods such as image inpainting (Balaji et al., 2022) aim to generate coherent parts of an image, while SDG (Liu et al., 2023) introduces semantic inputs to guide unconditional DDPM sampling. In addition, techniques such as (Meng et al., 2021) use images as editing conditions in denoising processes. Scene text generation methods such as TextDiffuser (Chen et al., 2023a;b) and GlyphDraw (Ma et al., 2023) have also emerged, utilizing textual layouts or masks to guide text generation in images. These advancements represent the growing versatility and potential of T2I models in diverse applications.

**Attention Guided Image Editing** has emerged as a fundamental solution to the challenge of translating human preferences and intentions into visual content through text descriptions. These approaches, as seen in works like (Li et al., 2023b; Avrahami et al., 2023; Zhang et al., 2023; Zhao et al., 2023), involve learning auxiliary modules on paired data. However, a limitation of these training-based methods is the substantial cost and effort required for repeated training for different control signals, model architectures, and checkpoints. In response to these challenges, training-free techniques have emerged, using the inherent weights of attention and the pre-trained models to control the attributes of objects such as size, shape, appearance, and location (Hertz et al., 2022; Epstein et al., 2023; Patashnik et al., 2023; Xie et al., 2023; Zhou et al., 2024a;b). These methods typically utilize basic conditions, such as bounding boxes, for precise control over object positioning and scene composition (Mo et al., 2023). Desigen (Weng et al., 2024) discovers relationship between attention and saliency and introduces attention reduction to weaken the attention within layout boxes. Our approach takes this further by applying force-directed graph techniques to cross-attention map edits, allowing for more automated and precise object transformations in T2I editing.

## 3. Method

Given an input text region $R$ and a prompt $T_{prompt}$, our framework aims to generate a text-friendly image $I_{res}$. This research is motivated by extensive interviews with wallpaper designers, who emphasized the need for precise control over text areas, object positioning, and background consistency in T2I-model-generated images. Specifically, our framework addresses these concerns by producing an

output image that has (1) reduced overlap between the primary object and $R$, (2) sufficient smoothness and minimal color variation in $R$, and (3) the image still fits the prompt. Our framework is shown in Figure 2.

As cross-attention map $A_k$ serves as a medium to locate and edit objects within generated images (Epstein et al., 2023), we focus on the $k^{th}$ token of $T_{prompt}$. In response to our first concern, we analyze the denoising process of an unguided image ($I_{ori}$) from a given prompt, and establish the subset of tokens ($O$) that refer to objects that need modification. We design a conflict detector that determines object conflicts based on the average attention intensity in the overlapping regions between $A_k$ and $R$. For every token $k \in O$, we introduce *Force-Directed Cross-Attention Guidance* for moving objects. In this scheme, objects are treated as centroid vertex within a graph, with a sequence of forces being applied (see Figure 3) to adjust object positions.

For the second consideration, inspired by recent technologies that limit the range of the attention map (Zhang et al., 2024), we propose a *spatial excluding cross-attention constraint* to prevent an extensive attention density from encroaching on $R$.

Addressing the third concern, our denoising process incorporates a loss function with additional regularization terms to safeguard the shapes and positions of other objects. Sometimes excessive repulsive force can occasionally displace essential objects from the image. To prevent objects from being dislocated outside limits while retaining their reasonable shapes, we also introduce the notions of *Margin Force $F_m()$* and *Warping Force $F_w()$*.

### 3.1. Force-Directed Cross-Attention Guidance

**Cross-Attention and Centroid of Object.** To seamlessly integrate the concept of force-directed graphs into the loss guidance of the denoising process in latent diffusion models, we delve into the extraction and manipulation of attention maps and activations. For denoising image $i$, we use softmax normalized attention matrices $\mathcal{A}_{i,t} \in \mathbb{R}^{H_i \times W_i \times K}$ extracted from the standard denoising forward step $\epsilon_{\theta_i}(z_i; t, y)$. This enables us to manipulate the control over objects referred to in the text conditioning $y$ at distinct indices $k$, by adjusting the related attention channel(s) $\mathcal{A}_{i,t,\dots,k} \in \mathbb{R}^{H_i \times W_i \times |k|}$. The centroid of the attention map is a two-dimensional vector, defined by the equation:

$$\text{centroid}(k) = \frac{1}{\sum_{h,w} \mathcal{A}_{h,w,k}} \begin{bmatrix} \sum_{h,w} h\mathcal{A}_{h,w,k} \\ \sum_{h,w} w\mathcal{A}_{h,w,k} \end{bmatrix}. \quad (1)$$

We assume that all objects are convex sets, adhering to the mathematical definition that for every pair of points within the object, the line segment connecting them lies entirely

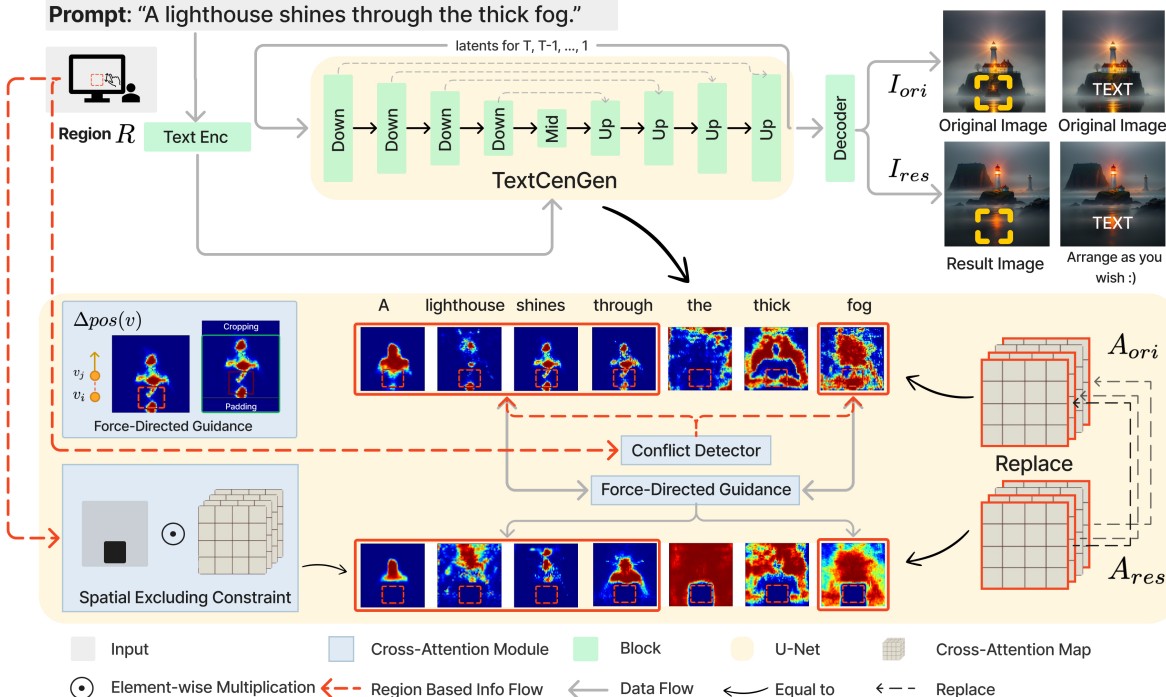

**Figure 2.** In our approach, the model receives a blank region ($R$) denoted as red-dotted area, and a text prompt as its inputs. The prompt is then used concurrently in a Text-to-Image (T2I) model to generate both an original image and a result image. During each step of the diffusion model's denoising process, the cross-attention map from the U-Net associated with the original image is used to direct the denoising of the result image in the form of a loss function. Throughout this procedure, a conflict detector identifies objects that could potentially conflict with $R$. To mitigate such conflicts, a force-directed graph method is applied to spatially repel these objects, ensuring that the area reserved for the text prompt remains unoccupied. To further enhance the smoothness of the attention mechanism, a spatial excluding cross-attention constraint is integrated into the cross-attention map.

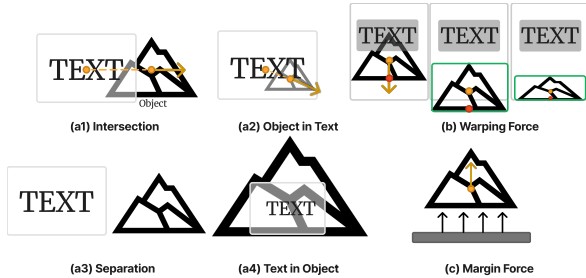

**Figure 3.** Illustration of four set relationships and their associated forces. The *Repulsive Force* separates object and text centroids during intersections (a1) and object in text (a2). The *Margin Force* (b) and *Warping Force* (c) prevent boundary overstepping. Text within object regions (a4) requires cooperation between force and attention constraint. Separation (a3) isn't required to process.

within the object. This assumption allows us to treat the extracted centroid $(k)$ as vertices $v_k$ in a graph, which are then subjected to force-directed attention guidance. Indeed, it is important to clarify that while each token $k$ associated with $\mathcal{A}_k$ appears as a single entity in Figure 2, it actually

represents an average of $\mathcal{A}_k^l$ across all layers $l$ in the U-Net architecture. Practically, our method is applied individually to each layer, ensuring a nuanced and layer-specific approach to force-directed attention guidance.

**Layer-wise Conflict Multi-Target Detector.** To effectively manage conflicts between text and objects in our images, we have developed a layer-wise conflict multi-target detector, denoted as $D()$. This detector is crucial for identifying tokens $k$ within each layer $l$ of the U-Net that correspond to objects that require modifications in relation to text regions. The detector function $D(k, R, A_k^l)$ operates as follows:

$$D(k, R, A_k^l) = \begin{cases} 1, & \text{if mean}(\mathcal{A}_{h,w,k}^l \cap R) > \theta \\ 0, & \text{otherwise} \end{cases} \quad (2)$$

where $\mathcal{A}_{h,w,k}^l$ is the attention map for token $k$ at layer $l$, and $R$ is the region designated for text. The function returns a value of 1 when the mean attention within the overlap between $\mathcal{A}_{h,w,k}^l$ and $R$ exceeds a predefined threshold $\theta$, indicating a conflict that requires our guidance function.

Identifying and adjusting the bounding boxes is visualized in Figure 2.

**Repulsive Force.** The fundamental repulsive force $F_{rep}(v_i, v_j) = \frac{-\xi^2}{||pos(v_i)-pos(v_j)||}$, ensures that each element is placed separately. $\xi$ denotes the general strength of the force, while $p(v_i)$ and $p(t/o)$ indicate the positions of the vertex and the target object, respectively. For scenarios that encompass multiple targets, our framework adopts a cumulative force approach to balance attention across these elements. This is quantified by the formula:

$$F_{mt}(v_i) = \sum_{j=1}^{n} \omega_j \cdot \frac{-\xi^2}{||p(v_i) - p(tar_j)||}, \qquad (3)$$

where $\omega_j$ are coefficients for balancing attention across targets. To regulate the impact of these forces and avoid excessive dominance by any single target, we introduce a force balance constant $\alpha$ in the form of $\frac{F_{rep}}{\alpha+F_{rep}}$. $\alpha$ ensures that the forces exerted do not exceed a practical threshold, thereby maintaining visual equilibrium in complex scenes.

**Margin Force.** The Margin Force is a critical component of our force-directed graph algorithm, designed to prevent significant vertices from being expelled from visual boundaries. This force, $F_m(v) = \frac{-m}{d(v,\text{border})^2}$, is activated as a vertex $v$ approaches the edge of the display area, typically a delineated rectangular space. The force is directed inward to ensure that crucial vertices remain within the designated visual region. The constant $m$ modulates the force's intensity, and $d(v, \text{border})$ represents the distance of the vertex $v$ from the nearest boundary (see Figure 3).

**Displacement and Position Update.** To compute the total displacement $\Delta pos(v)$, sum the repulsive and margin forces together: $\Delta pos(v) = F_{rep}(v) + F_m(v)$ Subsequently, update the vertex's position as follows: $pos(v) = pos(v) + \Delta pos(v)$. For the attention map $A_k^l$, this update is applied as a whole, with excess regions outside the boundaries being discarded and the remaining areas filled with zeros. But this method introduces the risk of the object being moved out of the boundaries and then being discarded, so it is necessary to introduce the following 'warping force' to prevent this from happening.

**Warping Force.** In addressing the dynamics of our force-directed graph algorithm, particularly for the movement of cross-attention maps $\mathcal{A}_k^l$ in each layer, we employ affine transformations as a pivotal mechanism. This approach facilitates the comprehensive translation and scaling of the entire map, preserving the relative positions within the image domain. Initially, we delineate our visual area or image space as a $H \times W$ two-dimensional array $A$. Within

this canvas $A$, we identify a key region, the object $O$, defined by the coordinates $(x, y, a, b)$, where $(x, y)$ marks the upper-left corner and $(a, b)$ the lower-right corner. The movement is calculated based on the sum of repulsive force $F_{\text{rep}}(v)$ and margin force $F_{\text{m}}(v)$, yielding the total displacement $\Delta \text{pos}(v) = F_{\text{rep}}(v) + F_{\text{m}}(v)$. Applying $\Delta \text{pos}(v)$ to both $A$ and $O$, we obtain the transformed canvas $A'$ and the object $O'$. Subsequently, we shift our coordinate system's origin to a vertex within $A'$ that remains within the canvas boundaries, establishing a new origin $O_{\text{new}}$. This repositioning is crucial when $O'$ exceeds the visual boundaries of $A$. In such cases, we scale the moved $\mathcal{A}_k^l$ to ensure that the bounding box of $O'$ fits precisely within the confines of $\mathcal{A}_k^l$. Scaling factors $S_x$ and $S_y$, are calculated as $S_x = \min\left(1, \frac{H-1}{a'}\right)$ and $S_y = \min\left(1, \frac{W-1}{b'}\right)$, where $(a', b')$ are the new coordinates of $O'$. Finally, the scaled $\mathcal{A}_k^l$ and $O'$ are reverted back to their original coordinate origin (see the warping force in Figure 3). A critical aspect of our approach is the transformation of the reference frame. After displacing $O$ due to $\Delta \text{pos}(v)$, a new reference frame is established, centered at $O_{\text{new}}$. The coordinates of $O$ in this new frame are calculated as $(x_{\text{new}}, y_{\text{new}}) = (x + \Delta x - O_{\text{new}_x}, y + \Delta y - O_{\text{new}_y})$. The affine transformation, considering this frame shift, is represented as:

$$\mathbf{T} = \begin{pmatrix} S_x & 0 & \Delta x - O_{\text{new}_x} \\ 0 & S_y & \Delta y - O_{\text{new}_y} \\ 0 & 0 & 1 \end{pmatrix}. \qquad (4)$$

This matrix transforms the coordinates of $O$, ensuring that it remains visible within $A$ after transformation. This carefully planned process secures the region $O$ within $A$, even after dynamic changes. It upholds the structure of the cross-attention map, balancing key vertices visibility and graph fluidity.

### 3.2. Spatial Excluding Cross-Attention Constraint

Our goal is to maintain a smooth background in the text region denoted as $R$. As illustrated in Figure 2, during each time-step of the forward pass in the diffusion model, we modify the cross-attention maps at every layer. The cross-attention map is represented as $\mathcal{A}_k^l \in \mathbb{R}^{H \times W \times K}$, where $H \times W$ are the dimensions of $A^l$ at different scales, and $K$ signifies the maximum token length at layer $l$. The set $I$ consists of indices of tokens corresponding to areas outside the text in the prompt. We resize $R$ to align with the $H \times W$ dimensions. Subsequently, a new cross-attention map for each layer $l$ is computed as $\mathcal{A}_{k,\text{new}}^l = \{\mathcal{A}_k^l \odot (1-R) \,|\, \forall k \in O\}$, where $O$ represents the tokens needing editing. This procedure effectively redirects the model's attention away from $R$, ensuring that the background in this region remains undisturbed and visually smooth. This spatially exclusive approach enhances the clarity and coherence of the

| Dataset | Metrics | Dall-E 3 | AnyText | Desigen | SD 1.5 | Ours (SD 1.5) | Improve (%) |
|---|---|---|---|---|---|---|---|
| P2P Template | Saliency IOU ↓ | 52.64 | 54.34 | 30.62 | 29.89 | **22.86** | 23.52% ↑ |
| | TV Loss ↓ | 18.02 | 22.55 | 13.7 | 14.11 | **8.81** | 37.56% ↑ |
| | VTCM ↑ | 1.92 | 1.75 | 2.92 | 2.95 | **4.4** | 49.15% ↑ |
| DiffuisonDB | Saliency IOU ↓ | 56.82 | 55.88 | 30.3 | 30.11 | **23.59** | 21.65% ↑ |
| | TV Loss ↓ | 21.16 | 20.39 | 11.99 | 12.3 | **8.41** | 31.63% ↑ |
| | VTCM ↑ | 1.67 | 1.79 | 3.20 | 3.19 | **4.39** | 37.62% ↑ |
| Syn Prompt | Saliency IOU ↓ | 51.52 | 53.24 | 31.57 | 31.42 | **27.7** | 11.84% ↑ |
| | TV Loss ↓ | 17.85 | 21.46 | 15.63 | 15.61 | **11.37** | 27.16% ↑ |
| | VTCM ↑ | 1.98 | 1.79 | 2.67 | 2.73 | **3.49** | 27.84% ↑ |

Table 1. Quantitative comparison of metrics across different methods and datasets. **Bold** indicate the best scores.

| Dataset | Metrics | SD 1.5 | Ours 1.5 | Imp (%) | SD 2.0 | Ours 2.0 | Imp (%) | SDXL | Ours XL | Imp (%) |
|---|---|---|---|---|---|---|---|---|---|---|
| P2P Template | Saliency IOU ↓ | 29.89 | **22.86** | 23.52 | 37.97 | 33.33 | 12.22 | 29.83 | 26.64 | 10.69 |
| | TV Loss ↓ | 14.11 | **8.81** | 37.56 | 17.73 | 12.06 | 31.98 | 12.09 | 9.10 | 24.73 |
| | VTCM ↑ | 2.95 | **4.40** | 49.15 | 2.52 | 3.38 | 34.13 | 3.41 | 4.24 | 24.34 |
| DiffusionDB | Saliency IOU ↓ | 30.11 | **23.59** | 21.65 | 33.40 | 29.52 | 11.62 | 34.22 | 32.31 | 3.16 |
| | TV Loss ↓ | 12.30 | **8.41** | 31.63 | 16.17 | 12.78 | 20.96 | 13.22 | 10.66 | 15.86 |
| | VTCM ↑ | 3.19 | **4.39** | 37.62 | 2.63 | 4.06 | 54.37 | 3.27 | 3.43 | 4.89 |
| Syn Prompt | Saliency IOU ↓ | 31.42 | **27.70** | 11.84 | 38.59 | 36.22 | 6.14 | 28.84 | 24.77 | 14.11 |
| | TV Loss ↓ | 15.61 | **11.37** | 27.16 | 18.92 | 15.25 | 19.40 | 12.31 | 8.48 | 31.11 |
| | VTCM ↑ | 2.73 | **3.49** | 27.84 | 2.42 | 2.83 | 16.94 | 3.59 | 4.78 | 33.15 |

Table 2. Performance comparison across different Stable Diffusion versions. **Bold** indicates the best scores for SD 1.5, which serves as our primary baseline. "Imp" represents the percentage improvement over the corresponding base model.

| Dataset | Dall-E 3 | AnyText | Desigen | **Ours** |
|---|---|---|---|---|
| P2P Template | 2.53 | 0.32 | 0.55 | **0.30** |
| DiffuisonDB | 2.04 | 1.04 | 0.34 | **0.31** |
| Syn Prompt | 2.25 | 1.02 | 0.51 | **0.29** |

Table 3. Quantitative comparison of CLIP Scores Loss for different methods. **Bold** indicates the best scores.

generated images, particularly in areas designated for text insertion.

# 4. Experiments

The evaluation is structured into quantitative and qualitative analyses, alongside an ablation study to understand the contribution of individual components of our model.

## 4.1. Implementation Details.

**Experimental Settings.** Our model is built with Diffusers. The pre-trained models are stable-diffusion-v1-5 and stable-diffusion-v2-0. While generating, the size of the output images is $512 \times 512$. We use one A6000 and ten A40 GPUs for evaluation. Detailed parameter settings are provided in the appendix.

**Dataset for Evaluation.** Our evaluation contains 27,000 images generated from 2,700 unique prompts, each tested in ten different random region $R$. The dataset combined synthesized prompts generated by ChatGPT, and 700 prompts from the Prompt2Prompt template (Hertz et al., 2022) designed for attention guidance, focusing on specific objects and their spatial relationships. Additionally, we included 1,000 DiffusionDB prompts (Wang et al., 2022), chosen for their real-world complexity. This diverse and comprehensive dataset, spanning synthetic to user-generated prompts, provided a broad test ground to evaluate the efficacy of our **TextCenGen** method in various T2I scenarios. Additionally, we constructed a targeted Desigen benchmark using 771 images from the Desigen dataset validation set (Weng et al., 2024), along with their corresponding static text masks, to evaluate performance on layout design-specific content. These 771 images were drawn from the original Desigen dataset of 53,577 usable images, where 52,806 were used for training the specialized Desigen model in Table 4.

## 4.2. Comparison with Existing Methods

We compared TextCenGen with several potential models to evaluate its efficiency. The baseline models included: Native Stable Diffusion (Rombach et al., 2022), Dall-E 3 (Ramesh et al., 2022), AnyText (Tuo et al., 2023) and Desigen (Weng et al., 2024). Dall-E used the prompt "*text-friendly in the {position}*" to specify the region $R$. Similar to AnyText, we chose to randomly generate several masks

| Dataset | Metrics | Designen-TF +AR | Designen-T +AR | **Ours (SD 1.5)** |
|---------|---------|-----------------|----------------|-------------------|
| P2P Template | Saliency IOU ↓ | 30.62 | 34.99 | **22.86** |
|  | TV Loss ↓ | 13.70 | 14.77 | **8.81** |
|  | VTCM ↑ | 2.92 | 2.97 | **4.40** |
| DiffusionDB | Saliency IOU ↓ | 30.30 | 31.11 | **23.59** |
|  | TV Loss ↓ | 11.99 | 14.58 | **8.41** |
|  | VTCM ↑ | 3.20 | 2.82 | **4.39** |
| Syn Prompt | Saliency IOU ↓ | 31.57 | 32.60 | **27.70** |
|  | TV Loss ↓ | 15.63 | 14.51 | **11.37** |
|  | VTCM ↑ | 2.67 | 2.93 | **3.49** |

*Table 4.* Comparison with trained versions of Designen on general datasets. Designen-TF: Training-free, Designen-T: Trained, AR: Attention Reduction. Even with specialized training on graphic design data, Designen-T does not match our training-free method's performance.

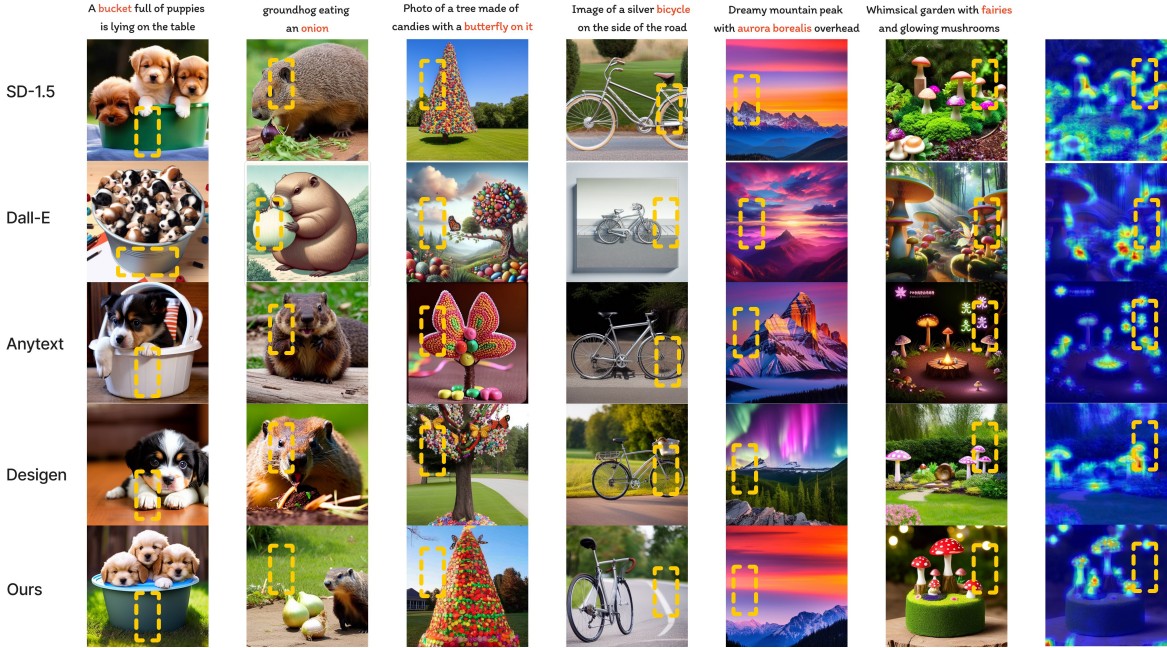

*Figure 4.* The results of comparison. Each column showcases six prompts across three datasets, the final column depicting the saliency map of the result image generated from the *mushroom* prompt. The red-dotted area denotes the planned blank region. Note that some methods fail to follow the orange-highlighted words in the prompt, leading to semantic loss.

| Method | Saliency IOU ↓ | TV Loss ↓ | VTCM ↑ |
|--------|----------------|-----------|--------|
| D-TF | 40.79 | 19.44 | 2.36 |
| D-T | 41.66 | 18.06 | 2.30 |
| **D-T+Ours** | 38.48 | 12.19 | 2.85 |
| **Ours (SD 1.5)** | **31.99** | **9.74** | **3.69** |

*Table 5.* Performance on the specialized Designen benchmark dataset. D-TF: Designen-Training-free, D-T: Designen-Trained. Underline indicates second-best performance.

in a fixed pattern across the map to simulate regions need to be edited. More detail can be found in appendix.

**Metrics and Quantitative Analysis.** To evaluate model performance, we used metrics assessing various aspects of the generated images. We proposed the CLIP Score Loss to evaluate the reduction in prompt semantic alignment for the training-free method compared to the vanilla diffusion model. The CLIP Score (Hessel et al., 2021; Huang et al., 2021; Radford et al., 2021) measured semantic fidelity, ensuring images align with textual descriptions. The total variation (TV) loss (Rudin & Osher, 1994; Jiang et al., 2021) assessed the visual coherence and smoothness of the background in relation to text region $R$, crucial for harmonious compositions. Saliency Map Intersection Over Union (IOU) (Qin et al., 2019) quantified the focus and clarity around text areas. We proposed the Visual-Textual Concordance Metric (VTCM), which combined a global metric increasing with value (CLIP Score) and local metrics that benefit from lower values (Saliency IOU and TV Loss)

| Dataset | Metrics | Text2Poster Best@5 | Text2Poster Avg@5 | **Ours (SD 1.5)** |
|---------|---------|--------------------|--------------------|-------------------|
| P2P Template | Saliency IOU ↓ | 31.25 | 36.22 | **22.86** |
| | TV Loss ↓ | 11.49 | 12.45 | **8.81** |
| | VTCM ↑ | 2.48 | 2.28 | **4.40** |
| | Clip Score ↑ | 20.80 | 20.80 | **27.96** |
| DiffusionDB | Saliency IOU ↓ | 24.07 | 34.38 | **23.59** |
| | TV Loss ↓ | **7.99** | 10.65 | 8.41 |
| | VTCM ↑ | 2.93 | 2.22 | **4.39** |
| | Clip Score ↑ | 17.57 | 17.57 | **27.20** |
| Syn Prompt | Saliency IOU ↓ | 31.25 | 36.91 | **27.70** |
| | TV Loss ↓ | 11.49 | 13.28 | **11.37** |
| | VTCM ↑ | 2.48 | 2.16 | **3.49** |
| | Clip Score ↑ | 20.80 | 20.90 | **28.10** |

*Table 6.* Comparison with retrieval-based methods. Text2Poster Best@5 shows the best results selected from five preset positions, while Avg@5 shows the average. Our method outperforms Text2Poster in most metrics, particularly CLIP Score, demonstrating better semantic fidelity while maintaining text-friendliness.

within $R$. The VTCM formula is:

$$\text{VTCM} = \frac{\text{CLIP Score}}{\text{Saliency IOU}} + \frac{\text{CLIP Score}}{\text{TV Loss}} \qquad (5)$$

Our quantitative analysis in Table 1 and Table 3 shows that **TextCenGen** minimizes semantic loss while maintaining smoothness and saliency harmony within region R, even surpassing the latest Dalle-3. Scene text rendering and attention reduction in Desigen often focus on local attention, neglecting global semantics. Especially in *Text in Object* situation (see Figure 3), this weakens the ability to create whitespace. Our approach moves main objects away, then reduces attention to create space, resulting in more natural and harmonious text layouts with the highest VTCM. The trade-off between background smoothness and semantic fidelity is shown in Figure 5.

We compared our method with the trained version of Desigen and retrieval-based methods like Text2Poster (Jin et al., 2022). Table 4 shows that the trained version of Desigen performs less effectively than our method despite using specialized graphic design data. As shown in Table 5, our plug-and-play method can be directly applied to pretrained model weights, yielding superior results compared to the same model using only attention reduction. When integrated with Desigen-Trained, our approach improves performance significantly, demonstrating that our training-free method can enhance specialized models without requiring additional training. As shown in Table 6, Text2Poster achieves competitive TV Loss on DiffusionDB, but our method provides better semantic fidelity (CLIP Score) and visual-textual coherence (VTCM). These results demonstrate the effectiveness of our training-free approach compared to both trained models and retrieval-based methods.

**MLLM-as-Judge ELO Ranking.** Following the rising trend of multimodal large language model (MLLM) as

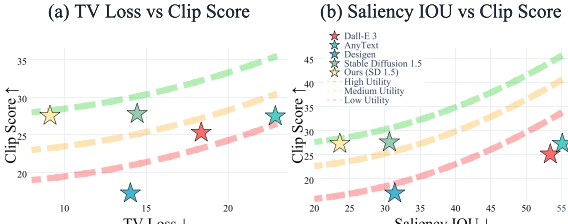

*Figure 5.* Performance trade-offs between different metrics. The dashed lines represent iso-utility curves, where points on the same curve indicate equivalent trade-off levels. Our method achieves a better balance between background smoothness and semantic fidelity. Higher utility curves (green) represent better overall performance.

judge methods (Chen et al., 2024; Wu et al., 2024), we present the MLLM-as-Judge ELO ranking for design appeal across different datasets in Table 7. The results demonstrate that the structured output from the GPT-4o provides consistent ratings, indicating that our method, along with Anytext and Dall-E, shows advantages in terms of design appeal. Interestingly, Dall-E excels in the Synthesized Prompts dataset, which contains a higher proportion of natural landscapes. For detailed information on the prompts and evaluation, please refer to the appendix.

| Rank | Method | DDB | P2P | SP |
|------|--------|-----|-----|-----|
| **1** | **TextCenGen** | **702.21** | **752.85** | **122.96** |
| 2 | Anytext | 279.56 | 329.32 | -89.68 |
| 3 | Dall-E | -17.32 | -39.05 | 78.87 |
| 4 | Desigen | -322.63 | -291.80 | -7.24 |
| 5 | SD1.5 | -629.33 | -738.83 | -92.40 |

*Table 7.* Method ELO design appealing rankings across three datasets.

**Qualitative Analysis.** Our qualitative analysis, shown in Figure 4, involved a comparison across different models using prompts and positions within the same quadrant. Dall-E 3, which relied solely on text inputs, exhibited significant variability and could not consistently clear the necessary areas for text placement. Desigen reduced attention in region $R$, but this method was not always effective without pre-trained graphic design-specific model weights, especially when region $R$ was within a main object, such as in the bicycle example. Introducing text to images was tricky, as TV Loss showed, but TextCenGen maintained object detail and background quality well. It showed that our force-directed method effectively balances text and visuals in images.

**User Study.** To understand the importance of the text-friendliness issue and explore users' subjective perceptions of the T2I Model's results, we conducted a user study with 114 participants. We used the qualitative results (see Figure 4) to develop a questionnaire. Figure 6 illustrates that the prompt-image alignment and aesthetics of our results were well-received in human perception.

### 4.3. Ablation Study

Table 8 presents the results of our ablation study. As part of our comprehensive analysis, we evaluated the two main contributors: (1) the impact of the force-directed component and (2) the effectiveness of the Spatial Excluding Cross-Attention Constraint.

**Impact of Force-Directed Cross-Attention Guidance.** The force-directed module is key to gently shifting where objects are placed. Without this, we might just bluntly edit the cross-attention map, which could mess up important parts of the picture. This part of our model helps us make sure we don't ruin the image structure by harshly removing attention map from areas.

**Effects of Spatial Excluding Cross-Attention Constraint.** Despite successful relocation of conflict object-related tokens, spaces left behind may not inherently lead to a well-blended transition. Our experimental results underscore that the integration of the Spatial Excluding Cross-Attention Constraint improve the smoothness of the remaining image sections.

## 5. Conclusion

We present TextCenGen, a plug-and-play method for text-friendly image generation and requires no additional training while well balancing both semantic fidelity and visual quality. This method abandons the traditional method of adapting text to pre-defined images. TextCenGen modifies images to adapt text, employing force-directed cross-

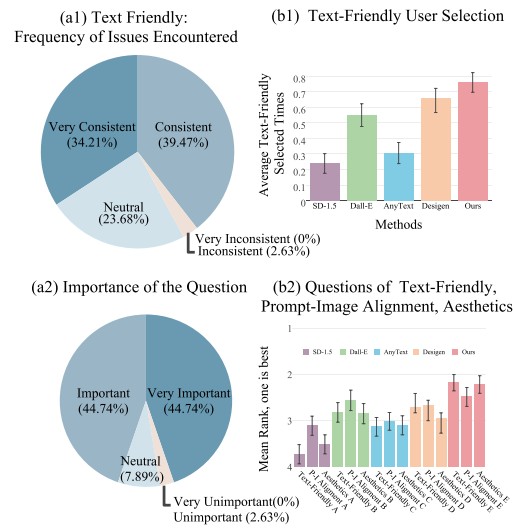

*Figure 6.* User study of task importance and result evaluation. The left shows user perceptions of task encounter frequency (a1) and importance (a2), rating 5 as the highest. The right side details user rankings (b2) and the average chosen times in a multiple-selection scenario (b1). The y-axis in b2 represents rankings from 1 to 5, demonstrating significant mean differences ($\alpha$=0.05) across three standards for all methods.

|  | w/o All | w/o FDG | w/o SEC | Ours |
|---|---|---|---|---|
| **CLIPS Loss** $\downarrow$ | - | 2.2 | 1.71 | **0.32** |
| **TV Loss** $\downarrow$ | 14.39 | 12.44 | 12.76 | **8.81** |
| **Saliency IOU** $\downarrow$ | 30.32 | 28.56 | 28.61 | **22.86** |
| **VTCM** $\uparrow$ | 2.93 | 3.03 | 3.05 | **4.4** |

*Table 8.* Ablation study results. We examine TextCenGen with or without the implementation of Force-Directed Cross-Attention Guidance (FDG) and Spatial Excluding Cross-Attention Constraint (SEC). The CLIP Score Loss of *w/o both* indicates the use of the vanilla SD-1.5 without our training-free method, resulting in no loss (-).

attention guidance to arrange whitespace. Furthermore, we have integrated a system to identify and relocate conflicting objects and a spatial exclusion cross-attention constraint for low saliency in whitespace areas.

Our approach has certain limitations. The force-directed cross-attention guidance, which assumes convexity and centering on object centroids, may not be suitable for non-convex shapes. This may lead to reduced size or fragmentation of objects. Future work will address these challenges to improve the quality of the output images.

## Acknowledgements

This work was supported by the NSSFC under Grant 22ZD05. We thank the anonymous reviewers and Dr. Sicheng Song for their valuable feedback and suggestions.

## Impact Statement

This paper presents work whose goal is to advance the field of Machine Learning. There are many potential societal consequences of our work, none of which we feel must be specifically highlighted here.

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

We provide more details of the proposed method and additional experimental results to help better understand our paper.
**Contents**

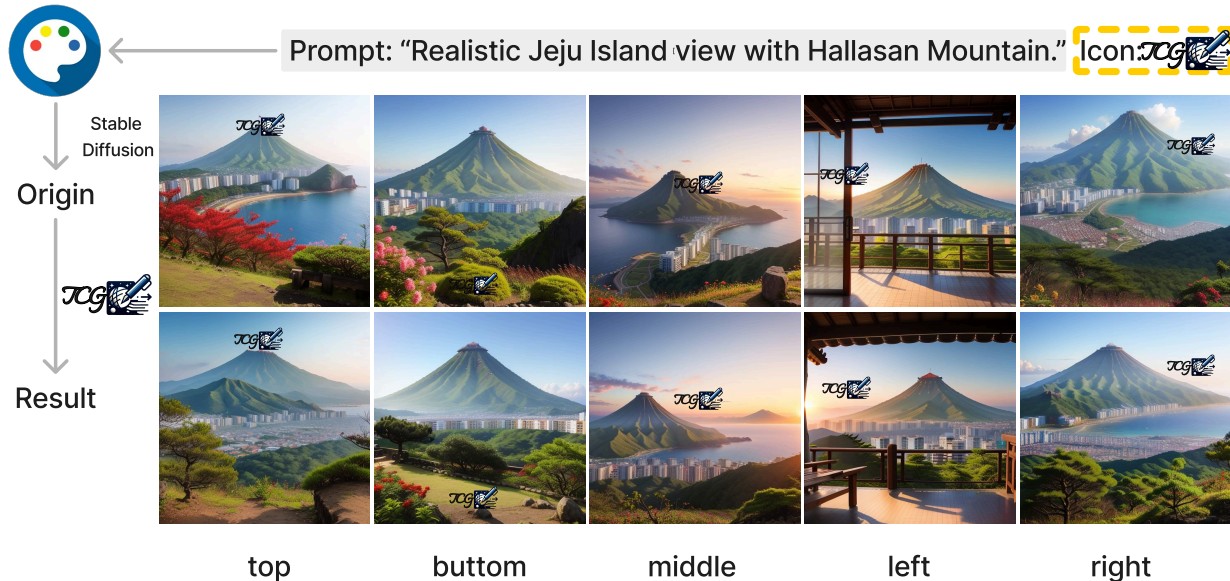

Figure 7. **Can you spot the TCG logo at first glance?** TextCenGen is a training-free method designed to generate text-friendly images. It simultaneously generates the original and result images. During the denoising process of the original image, the generation of the result image is guided based on the encroachment into the planned blank regions in the original. This approach ensures sufficient blank space at specific positions, typically where text or icons are centered, in the result image.

## A. Task Introduction

Text-friendly images refer to images designed or selected with an emphasis on enhancing the readability and clarity of overlaid text. These images typically have simple, non-distracting backgrounds, a balanced color palette, and areas of

negative space that can accommodate text without compromising visibility or design aesthetics. Common applications include marketing materials, presentations, and social media graphics where the text plays a crucial role in conveying the message. Key considerations for creating or selecting text-friendly images include contrast, alignment, and ensuring the image content does not compete with the overlaid text.

| Task Name | Don't Requires Training | Don't Requires Annotation | Type of Layout Specification | Required Number of Anchors |
|---|---|---|---|---|
| Layout-to-Image (Rombach et al., 2022) | X | X | Object | > 5 |
| Text-Friendly Image Generation | ✓ | ✓ | Space Region | 1-2 |
| Visual Text Generation | ✓ | ✓ | Text | 1-2 |

*Table 9.* Comparison of our task with existing tasks. Unlike layout-to-image tasks requiring training and intensive annotation, our method only needs space region annotation as input for downstream modifications. This makes it particularly suitable for applications such as dynamic wallpapers for mobile devices and e-commerce posters.

## B. Experiment Setting

Our proposed model is designed using the Diffusers library, specifically leveraging the stable-diffusion-v1-5 pre-trained models with DDPM Scheduler. The model generates images of dimensions $512 \times 512$. In our method, we have set the force balance constant $\alpha$ to 0.5. The coefficients for regularization term $\gamma$ is fixed at 0.01. Within the detector, we upscale all cross-attention maps to a $64 \times 64$ resolution. Additionally, we expand the height and width of the region $R$ by a margin of 0.06. During the first 20 steps, we identify conflicting objects when the Intersection over Union (IOU) exceeds 0.14. For the subsequent 30 steps, we initiate a push operation only if the average density inside region $R$ surpasses 0.8. Negative prompts are *"monocolor, monotony, cartoon style, many texts, pure cloud, pure sea, extra texts, texts, monochrome, flattened, lowres, longbody, bad anatomy, bad hands, missing fingers, extra digit, fewer digits, cropped, worst quality, low quality"*.

Our cross-attention replacement method requires less than 15GB of VRAM, making it feasible to run inference on consumer GPUs like the RTX 3090. For evaluation purposes, we utilized one NVIDIA A6000 and 8 H800 GPUs. The entire experimental assessment took approximately 96 hours to complete. Particularly on the A40 GPU, the image generation process, which includes both the original and the resultant images, took around 50 seconds for 50 steps of inference. In contrast, utilizing attention guidance with difftext results in a faster average inference time of approximately 30 seconds, as it only requires the inference of a single image.

### B.1. Region Random Sampling Method

Our region random sampling method is a variation inspired by DiffText. It involves two predefined regions, measuring $160 \times 64$ and $64 \times 160$. During the evaluation, excluding Dall-E, we randomly select areas of these dimensions from the entire image. The output image of Dall-E is not $512 \times 512$, so we proportionally scale down the corresponding regions to $116 \times 46$ and $46 \times 116$ while calculating metrics. Given that Dall-E operates exclusively on textual inputs as provided by the prompt, we intend to specify regions within Dall-E's output at five distinct locations: left, right, bottom, top, and center.

### B.2. Analysis of Text Box Shape Orientations

The shape orientation of text boxes was randomly generated following our region random sampling method, ensuring an unbiased experimental setup. To investigate potential biases related to text box orientations, we conducted additional analysis by separating results based on horizontal ($160 \times 64$) and vertical ($64 \times 160$) orientations. Table 10 shows the differences between horizontal and vertical orientations across different metrics and datasets, calculated as horizontal minus vertical values.

TextCenGen shows relatively consistent performance across orientations, with slightly better metrics for horizontal boxes in the P2P Template dataset. Different methods exhibit varying preferences for orientation, suggesting the underlying complexity of text-friendly image generation across different box shapes. The variance in performance across orientations is generally smaller for TextCenGen compared to baseline methods, indicating more robust handling of different text box

| Dataset | Metrics | SD | Dall-E 3 | AnyText | Desigen | **Ours** |
|---|---|---|---|---|---|---|
| | Saliency IOU | -1.54 | 0.17 | -1.17 | -0.57 | 0.31 |
| P2P Template | TV Loss | -4.83 | 5.89 | -9.77 | -1.26 | 3.77 |
| | VTCM | -0.42 | 0.00 | 0.00 | 0.09 | 0.55 |
| | Saliency IOU | 0.83 | -0.94 | 0.00 | 0.32 | 0.00 |
| DiffusionDB | TV Loss | 2.57 | 2.00 | -3.48 | 2.49 | -0.66 |
| | VTCM | -0.32 | 1.76 | -0.12 | 0.09 | -1.14 |

*Table 10.* Differences between horizontal and vertical text box orientations across metrics and datasets. Positive values indicate better performance with horizontal boxes.

shapes.

### B.3. Evaluation Metrics

To evaluate model performance, we used 4 metrics assessing various aspects of the generated images. The CLIP Score, total variation (TV) loss, Saliency Map Intersection Over Union (IOU) and Visual-Textual Concordance Metric (VTCM).

- **CLIP Score** is described as a metric used to measure the similarity between generated images and input prompts, employing off-the-shelf code referenced as (cli, 2022). Typically, the original image achieves the highest CLIP score when it's associated with stable diffusion techniques. However, when blank areas are defined within an image, there might be a slight reduction in the CLIP Score. This score is crucial in determining how closely an AI-generated image aligns with the given textual prompt, playing a significant role in the overall evaluation of the image's fidelity to the intended text description.

- **Total Variation Loss** is a regularisation term commonly used in image processing tasks, particularly those involving image reconstruction or denoising. It is designed to encourage spatial smoothness in the output image while preserving important structural details. The total variation loss for a region $R$ can be computed as follows:

$$\text{TV}(R) = \sum_{i,j \in R} \sqrt{(\Delta_x R_{i,j})^2 + (\Delta_y R_{i,j})^2}$$

Here, $\Delta_x R_{i,j}$ and $\Delta_y R_{i,j}$ represent the discrete differences in the horizontal (x-axis) and vertical (y-axis) directions, respectively, at pixel location $(i, j)$ within the region $R$. The term $(\Delta_x R_{i,j})^2 + (\Delta_y R_{i,j})^2$ calculates the squared gradient magnitude at each pixel, and the summation is taken over all pixels within the region $R$.

This formulation of the total variation loss ensures that the reconstructed or processed region $R$ does not have abrupt changes in pixel values, leading to a smoother and more visually appealing result.

- **Saliency Map Intersection Over Union** is specifically tailored for assessing the overlap between a saliency map and a designated region within an image. The formula for this metric is given as:

$$\text{IoU}(S, R) = \frac{|S \cap R|}{|S \cup R|}$$

In this context, $S$ represents the saliency map, and $R$ denotes a specific region in the image. The term $|S \cap R|$ is the count of pixels that are common to both the saliency map $S$ and the region $R$, signifying their intersection. On the other hand, $|S \cup R|$ refers to the count of pixels present in either the saliency map $S$ or the region $R$, indicating their union. In our case, a lower IoU value is desirable as it indicates less overlap, aligning with our goal to ensure that the ROI is non-salient and distinct from the saliency map.

- **Visual-Textual Concordance Metric** is formulated to assess the coherence between text prompts and generated images in AI-driven text-to-image synthesis. Defined as $\text{VTCM} = \text{CLIP Score} \times \left( \frac{1}{\text{Saliency IOU}} + \frac{1}{\text{TV Loss}} \right)$, it combines three elements. The CLIP Score reflects the degree of match between the generated image and the text prompt, where higher scores indicate better alignment. The Saliency IOU (Intersection Over Union) in Region R measures how well the most salient parts of the image align with the specified region, with lower scores being better. The TV Loss in

Region R assesses the smoothness or consistency in the image's region, where lower TV Loss scores indicate a more uniform and less noisy region. The VTCM thus encourages the generation of images that are not only coherent with the text prompt but also exhibit focused quality in specified areas, aligning with the goal of creating text-friendly images.

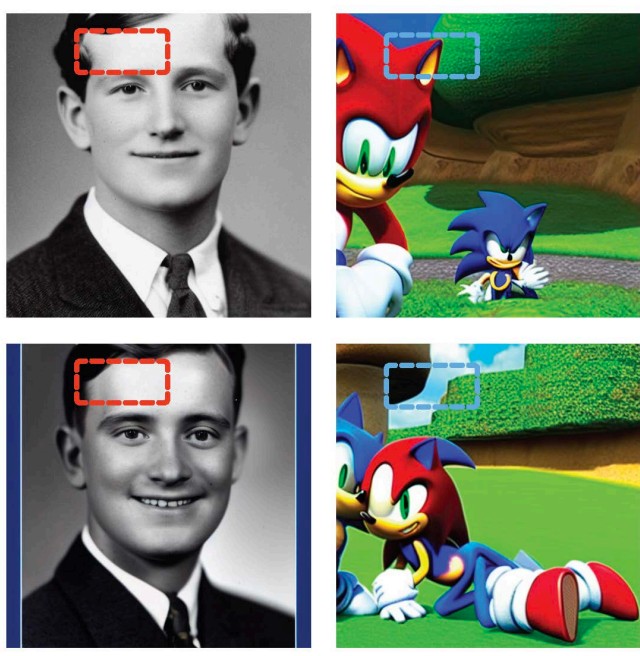

*Figure 8.* The Limitation of Our Model.

A playful dolphin jumping out of the ocean, text-friendly in the {position}

image:

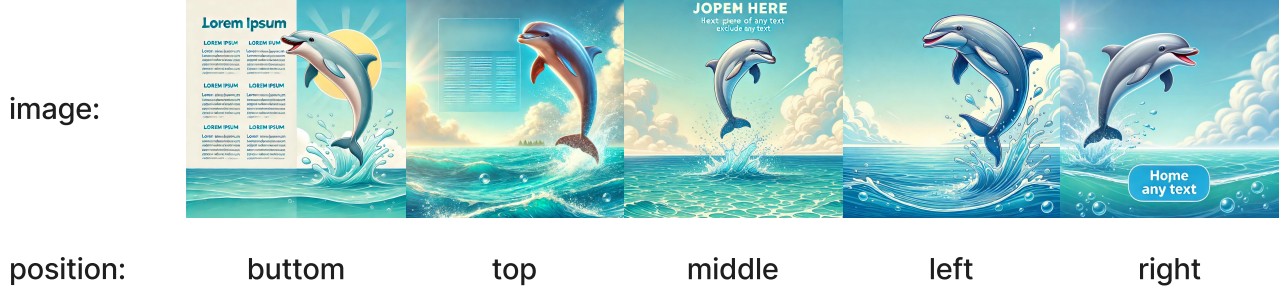

position:      buttom        top        middle        left        right

*Figure 9.* The data generation method in Dall-E.

## B.4. Details of Compared Methods

We compare the proposed model with Stable Diffusion, Dall-E, AnyText and DiffText. The details of compared methods are as follows:

- **Stable Diffusion**, as detailed in Rombach et al. (2022) (Rombach et al., 2022), is an innovative open-source model. We utilize the widely accessible pre-trained model labeled as "runwayml/stable-diffusion-v1-5". For our experiments, we set the sampling steps to 50 and the classifier-free guidance scale at 7.5. The model serves both as the source for generating attention guidance from the origin image in our methodology and as one of the compared methods.

- **Dall-E** (Ramesh et al., 2022) is a groundbreaking AI model developed by OpenAI, known for its capability to generate complex images from textual descriptions. Utilizing a variant of the GPT-3 architecture, Dall-E transforms text inputs

into detailed and creative visual outputs, showcasing a deep understanding of both language and visual concepts. In our experiments, the output image of Dall-E is not $512 \times 512$, so we proportionally scale down the corresponding regions to $116 \times 46$ and $46 \times 116$ while calculating metrics. To enhance the focus on text compatibility at these positions, we add the phrase 'text-friendly in the corresponding position' directly into the original prompts, as depicted in Figure 9. Restricted to the purely textual input of Dall-E, the performance falters when the position falls between left and right and when the text lacks precision.

- **AnyText** (Tuo et al., 2023) is a diffusion-based multilingual visual text generation and editing model that focuses on rendering accurate and coherent text in the image that outperformed all other approaches. We generate a fully black mask for that region and fully white for the rest of the image. Then we use this mask to replace the "draw_pos" in the input parameters, while keeping the other parameters unchanged.

- **Desigen** (Weng et al., 2024) is an automatic template creation pipeline which generates background images as well as harmonious layout elements over the background as well as an iterative inference strategy to adjust the synthesized background and layout in multiple rounds is presented. For fair comparison, we adopted the following approaches: (1) We used random regions as layout elements, similar to other baselines. (2) Our comparison targets are training-free methods, so we used vanilla SD1.5 without any LoRA as the base weights. (3) Using Desigen's attention reduction method on this base weight, we set the attention ratio within the region to 0.

### B.5. MLLM-as-Judge ELO Ranking

To comprehensively assess the performance of TextCenGen against other baseline methods, we employed the Elo rating system, a method originally developed for ranking chess players but now widely used in various competitive contexts (Duan et al., 2024; Chen et al., 2024; Wu et al., 2024). In our evaluation framework, we apply a multi-modal large language model (MLLM) as a judge to compare TextCenGen with other baseline methods. The evaluation is based on the Elo rating system, commonly used in competitive ranking, which allows for continuous adjustments of scores as pairwise comparisons are made. Specifically, for each comparison between two methods, the MLLM assesses the "Design Appeal" of the generated outputs and determines a winner. The Elo method then updates the ratings of the two competing methods accordingly.

The Elo rating process involves calculating the expected scores $E_A$ and $E_B$ for methods $A$ and $B$, given their current ratings $R_A$ and $R_B$. The expected scores are computed using the formula:

$$E_A = \frac{1}{1 + 10^{\frac{R_B - R_A}{400}}}, \qquad E_B = \frac{1}{1 + 10^{\frac{R_A - R_B}{400}}} \tag{6}$$

Based on the comparison outcome $S_A$ (where $S_A = 1$ if method $A$ wins, or $S_A = 0$ if it loses), the ratings are updated using:

$$R'_A = R_A + K \times (S_A - E_A)$$
$$R'_B = R_B + K \times ((1 - S_A) - E_B)$$

Here, $K$ represents the adjustment factor, set to 32 in our experiments, which determines the sensitivity of rating changes.

In practice, our evaluation system iterates through each dataset, initializing the Elo scores for all methods and adjusting them dynamically as new comparison results are processed. After all comparisons, the methods are ranked based on their aggregated Elo scores, providing insights into the relative strengths of TextCenGen and other baselines in terms of "Design Appeal." This approach ensures a consistent and scalable evaluation across diverse datasets while reflecting the preferences derived from the MLLM's judgments.

## C. Influence of the force balance constant

$\alpha$ ensures that the forces exerted do not exceed a practical threshold, thereby maintaining visual equilibrium in complex scenes (shown in Figure 12).

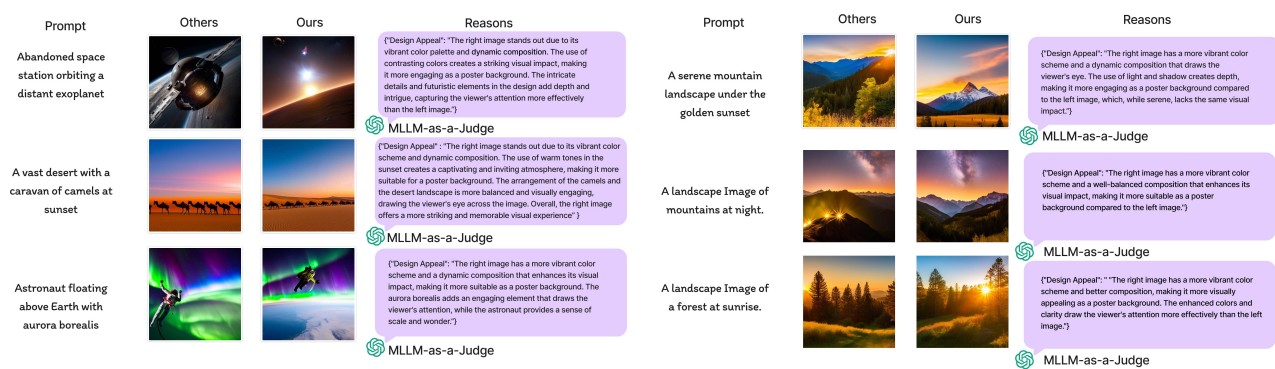

*Figure 10.* More results of elo mllm judge.

## D. More Results of Proposed Method

We present additional outcomes of the proposed methodology. Referring to Figure 11, our model produces 12 instances drawn from three distinct datasets. The figure also illustrates the strong performance of our approach across diverse scenarios. These encompass urban landscapes, natural scenes, still life, Valentine's Day greeting cards, and environments from both 3D and 2D video games.

### D.1. More Result of Ablation Study

Figure 13 presents the results of our ablation study. This study was designed to elucidate the individual contributions of the various components embedded within our proposed model. As part of our comprehensive analysis, we evaluated the three main contributors: (1) the impact of the force-directed component, (2) the effectiveness of the Spatial Excluding Cross-Attention Constraint.

### D.2. Compatible with Lora Checkpoint

Figure 1 demonstrates the effect of using LoRA by utilizing the LoRA rev-animated model from Civitai for an animated style, while the qualitative evaluation uses original SD weights for comparison.

## E. Limitations of Our Model

Our approach could result in unexpected changes, exemplified by the left of Figure 8, where one might initially think the figure shifts downward, rather than the forehead area. We noted the emergence of unintended objects within the empty spaces on the right of Figure 8, spaces which the original prompts did not specify.

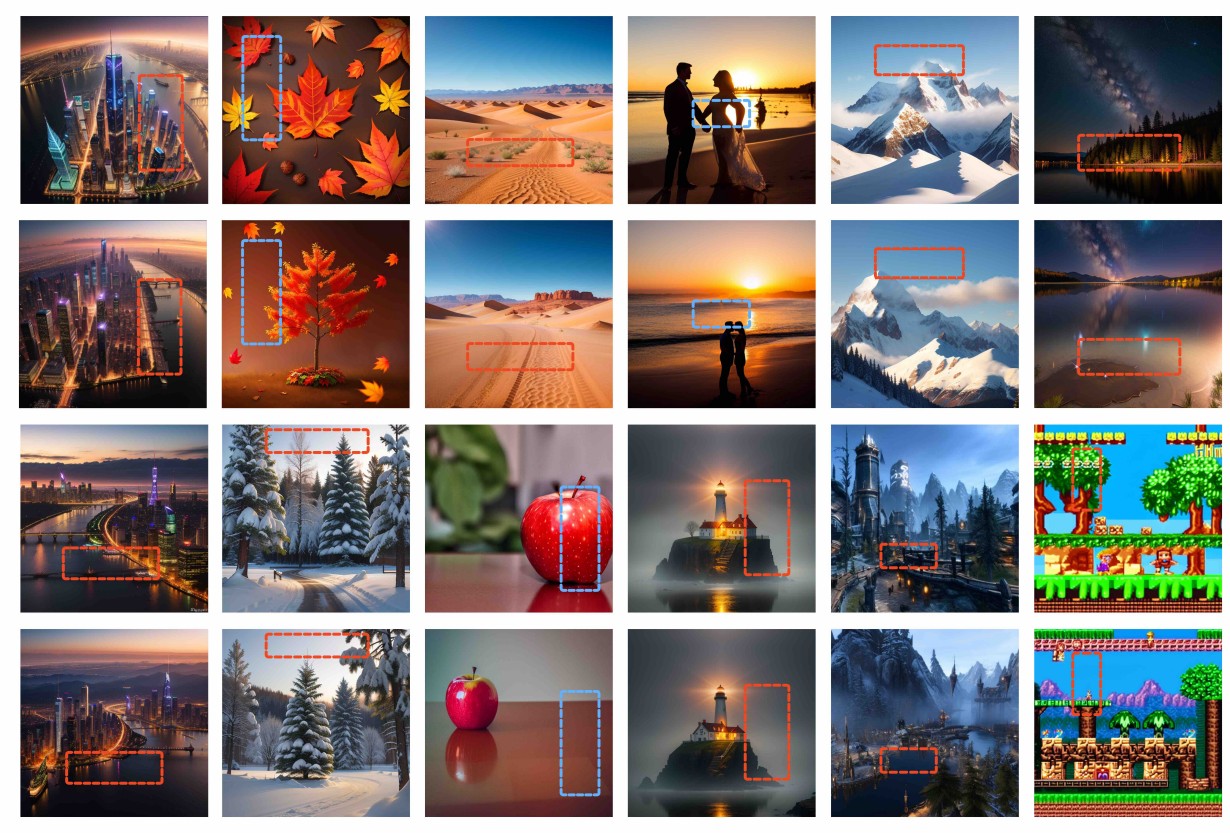

*Figure 11.* More results of the proposed method. The first and third lines display the original images, while the second and fourth lines exhibit the result images.

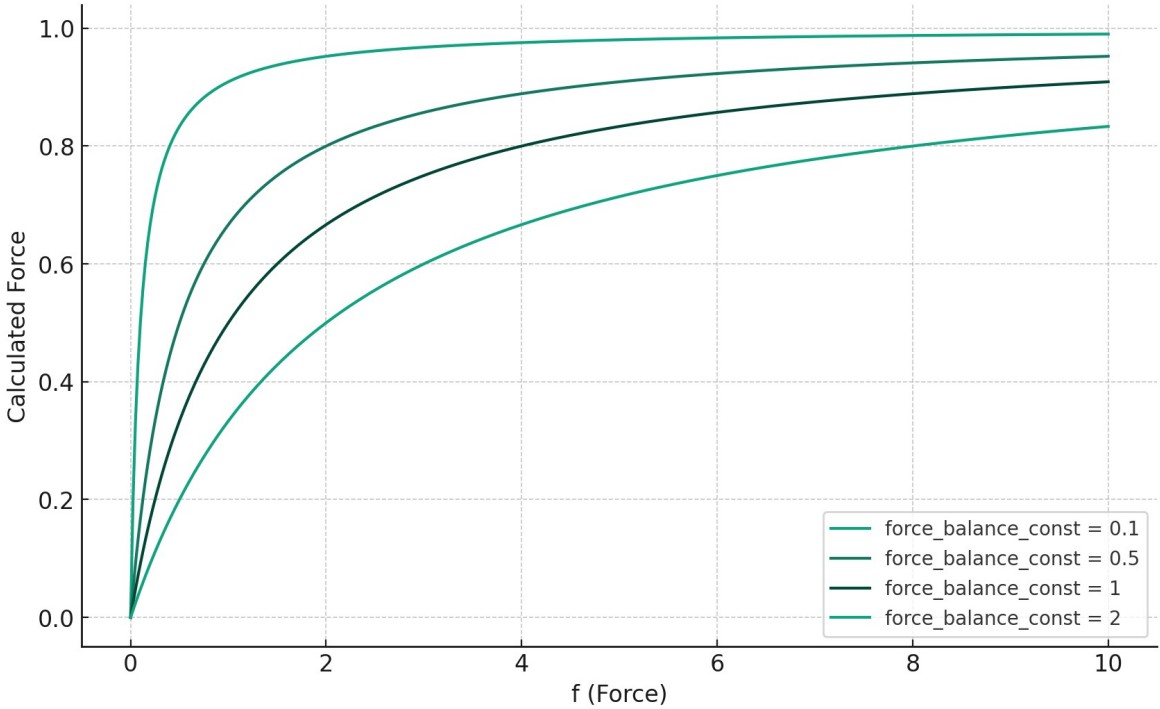

*Figure 12.* Influence of the force balance constant. The graph illustrates the effects of four different values for $\alpha$ on $F_{rep}$ versus the final force. The curves reveal that as $\alpha$ increases, the rate of convergence to the limit progresses more rapidly.

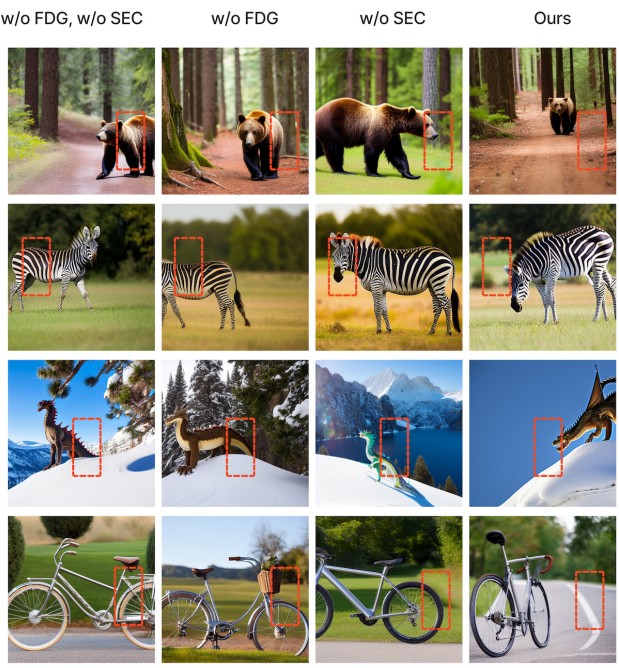

*Figure 13.* The results of ablation. We examine TextCenGen with or without the implementation of Force-Directed Cross-Attention Guidance (FDG) and Spatial Excluding Cross-Attention Constraint (SEC). The red-dotted area denotes the target area preserved for either text or icon images. Images produced with both FDG and SEC yield outcomes identical to those created using TextCenGen alone. Conversely, images created without FDG and SEC are comparable to those derived from the original stable diffusion model.

