# OpenReview forum: "TextCenGen: Attention-Guided Text-Centric Background Adaptation for Text-to-Image Generation"
_ICML.cc/2025/Conference — ICML 2025 poster_

### Official Review · Reviewer_XmgK · 2025-03-05

**Overall Recommendation:** 3

**Summary:**

This paper mainly targets text-to-image generation. The authors focus on an interesting problem: after generating images, one would potentially want to insert specific visual texts into the images, and it would be better if the area to be inserted has clean background rather than being occupied with other objects. The authors show that current T2I models can hardly fulfill this goal without specific design. To this end, they propose an attention-guided text-centric background adaptation to modify the attention maps according to several criteria. The authors conduct multiple experiments to show the effectiveness.

**Claims And Evidence:**

Yes, the authors have pointed out an interesting and important problem in T2I generation. The experiments can validate this.

**Essential References Not Discussed:**

NA

**Experimental Designs Or Analyses:**

Yes. The main experiment includes comparison with several T2I models. Both qualitative and quantitative results illustrate the superiority of the proposed method. The ablation study can show the role of each term in the proposed guidance.

**Methods And Evaluation Criteria:**

Yes, the relationship between attention map and generated objects has been studied in previous works. Besides, the experiments designed by the authors are reasonable enough to show the effectiveness of the proposed method.

**Other Comments Or Suggestions:**

Please refer to the weaknesses.

**Other Strengths And Weaknesses:**

Strengths:
1. The whole paper is well written and easy to understand.

Weaknesses:
1. I wonder if it would be better to place the task importance in Fig.6 in Sec.1?
2. It has been shown that actually models such as SD1.5 and SDXL do not enjoy strong correlation between the attention map and the target objects. I wonder if this could affect the proposed method.
3. While the proposed method is solid, it is to some extent complicated. Some simpler methods, for example, generating staggered layouts and using layout-grounded T2I to generate images, or directly restricting the TV in the target region, may be a better choice.
4. It would be better to present results with more advanced diffusion models such as SD3 and FLUX.

**Questions For Authors:**

Please refer to the weaknesses.

**Relation To Broader Scientific Literature:**

The proposed method can better enhance the application of T2I models in design of wall papers, posters, etc. Besides, the method can be potentially extended to T2V/T-2-3D models.

**Theoretical Claims:**

No theoretical claims are presented in this paper.

---

> ### Author Rebuttal · Authors · 2025-03-30
>
> Thank you for your valuable review. We address each point below.
>
> ### [W1]Task Importance Placement in Introduction
>
> We'll add to the introduction: "As shown in Fig. 6, creating text-friendly images is essential for graphic design applications (validated by our 114-participant user study)."
>
> ### [W2]Correlation Between Attention Maps and Target Objects
>
> Correlation is an important assumaption for our method, which relies on attention manipulation. Strengthening the correlation between semantic content and attention maps is a critical research direction in this field [1,2]. **Any training-free methods that improve this correlation could be readily integrated as upstream components to our approach.** This represents a promising direction for future research that would complement rather than replace our contribution. Notably, Our results still demonstrate that attention manipulation remains an effective mechanism for our task even with current models with semantic loss (with clip score).
>
> ### [W3]Alternative Approaches Comparison
>
> To clarify differences between our task and existing approaches, we present the following comparison:
>
> | Task | Training-Free | Annotation-Free | Layout Specification | Required Anchors |
> |------|---------------|-----------------|----------------------|------------------|
> | Layout-to-Image [3] | ❌ | ❌ | Object | >5 |
> | Text-Friendly Image Gen (Ours) | ✓ | ✓ | Space Region | 1-2 |
> | Visual Text Generation | ✓ | ✓ | Text | 1-2 |
>
> Unlike layout-to-image tasks requiring training and intensive annotation, our method only needs space region specification, which is crucial for dynamic applications like mobile wallpapers (Figure 1) or e-commerce posters.
>
> Layout-based methods primarily focus on object arrangement rather than creating text-friendly backgrounds. Their objective is fundamentally different:
> - Layout-grounded T2I: Positions objects according to a layout
> - Our approach: Creates backgrounds that harmonize with planned text regions
>
> Direct TV restriction in target regions would yield effects **similar to attention reduction**, which we have already validated in our ablation studies. As shown in Table 1, simply applying spatial constraints without force-directed guidance (w/o FDG) leads to higher CLIPS Loss (2.2 vs 0.32), indicating reduced semantic fidelity to the original prompt. This demonstrates that naive approaches like direct TV restriction fail to balance text-friendliness with instruction following.
>
> | Method | CLIPS Loss ↓ | TV Loss ↓ | Saliency IOU ↓ | VTCM ↑ |
> |--------|-------------|-----------|----------------|--------|
> | w/o FDG | 2.2 | 12.44 | 28.56 | 3.03 |
> | Ours | **0.32** | **8.81** | **22.86** | **4.4** |
>
> Our force-directed approach enables precise object placement control while maintaining semantic coherence, as evidenced by lower CLIPS Loss and superior text-compatibility metrics.
>
>
> **Both alternatives would ultimately need similar attention control mechanisms to achieve our goal of text-friendly background generation that respects user-specified regions**.
>
> ### [W4]Testing with More Advanced Models
>
> We have deployed our method on SD1.5, SD2.0, and SDXL to demonstrate the broad applicability of our method across different model architectures. Our approach can be adapted to MMDiT-based models like FLUX through methods similar to those demonstrated in recent work [4] similar to this [code](https://github1s.com/krafton-ai/Rare-to-Frequent/blob/main/R2Fplus_Diffusion_sd3.py#L986). The key difference in MMDiT models is the attention structure, which can be described as:
> ```
> MMDiT Attention =  | Text-Text (TT) | Text-Image (TI) |
>                    | Image-Text (IT) | Image-Image (II)| ,
> ```
>
> In UNet models, we modify cross-attention (equivalent to TI block). For MMDiT models, one possible solution is to apply our force-directed approach to the Image-Text (IT) block (such as transpose the IT block), and maintain consistency in the Text-Image (TI) block.
>
> This adaptation is technically straightforward based on existing implementations like Rare-to-Frequent. While implementation details differ, the fundamental principle of manipulating attention to create text-friendly backgrounds remains valid across architectures.
>
> We appreciate your thoughtful suggestions and will address them in our revised manuscript.
>
> [1] Yang et al. "Dynamic prompt learning: Addressing cross-attention leakage for text-based image editing," Neurips 2023
>
> [2] Liu B, Wang C, Cao T, et al. "Towards understanding cross and self-attention in stable diffusion for text-guided image editing," CVPR 2024
>
> [3] Zheng et al., "LayoutDiffusion: Controllable diffusion model for layout-to-image generation," CVPR 2023
>
> [4] Park D, Kim S, Moon T, et al. "Rare-to-Frequent: Unlocking Compositional Generation Power of Diffusion Models on Rare Concepts with LLM Guidance," ICLR 2025 Spotlight

---

### Official Review · Reviewer_tH44 · 2025-03-10

**Overall Recommendation:** 4

**Summary:**

This paper introduces TextCenGen, a training-free method for generating text-friendly images. While traditional text-to-image (T2I) models can create high-quality images, they typically don't account for the need to reserve space for text placement. TextCenGen addresses this challenge through several innovations:

1) A dynamic background adaptation method that creates smooth backgrounds for predefined text regions
2) A force-directed graph algorithm that guides the relocation of objects in attention maps to avoid overlap with text regions
3) A spatial excluding cross-attention constraint that ensures background smoothness in text regions

The authors constructed a benchmark with 27,000 images, demonstrating that TextCenGen reduces saliency overlap in text regions by 23% compared to existing methods while maintaining 98% of the original semantic fidelity. The method requires no additional training and can be plugged into various T2I models.

**Claims And Evidence:**

The paper's claims are well-supported by experimental evidence:

1. The authors propose new evaluation metrics (Saliency IOU, TV Loss, CLIP Score Loss, and VTCM) that comprehensively assess the quality and text-friendliness of generated images.
2. Quantitative comparisons with existing methods (Dall-E 3, AnyText, Desigen, and Stable Diffusion 1.5) show that TextCenGen outperforms these methods across all datasets.
3. A user study with 114 participants validates the effectiveness and user perception of the method.
4. Detailed ablation studies demonstrate the importance of force-directed cross-attention guidance and spatial excluding cross-attention constraint.
5. Rich visual results intuitively showcase the method's effectiveness.

**Essential References Not Discussed:**

The paper already includes the major literature in related fields

**Experimental Designs Or Analyses:**

The experimental design is comprehensive and reasonable:

1. Benchmark comparison: Thorough comparison with existing methods across multiple datasets
2. Ablation studies: Evaluation of the contribution of each component
3. User study: Validation of the method's practicality and user perception

The experimental analysis is also thorough:
1. Combines quantitative metrics and qualitative results
2. Compares performance across different Stable Diffusion versions
3. Analyzes the influence of the force balance constant
4. Provides generalization analysis across different types of prompts and region shapes

**Methods And Evaluation Criteria:**

The proposed methods are well-suited for addressing the text-friendly image generation problem:

1. Force-Directed Cross-Attention Guidance: Cleverly applies physical mechanics concepts to attention map editing, enabling smooth relocation of conflicting objects.
2. Spatial Excluding Cross-Attention Constraint: Ensures background smoothness in text regions, suitable for text overlay.
3. The training-free approach makes the method more accessible and applicable to various pre-trained models.

The evaluation criteria are comprehensive and reasonable:
1. Testing on three datasets (P2P Template, DiffusionDB, and Synthesized Prompts) covers diverse scenarios.
2. Evaluation metrics balance considerations of image quality, semantic fidelity, and text region adaptability.
3. MLLM-judged ELO ranking provides a more comprehensive assessment of design appeal.

**Other Comments Or Suggestions:**

1. Consider providing a simplified algorithm pseudocode to make the entire process easier for readers to understand
2. Further testing in practical application scenarios (such as poster design, mobile interface design) would strengthen the work
3. Provide more analysis on computational complexity and runtime

**Other Strengths And Weaknesses:**

Strengths:
1. Addresses an important and practical problem: generating images with space reserved for text
2. The training-free method design is clever and easy to integrate into existing models
3. Comprehensive evaluation including multiple datasets, metrics, and user studies
4. The proposed force-directed cross-attention guidance concept is innovative, providing a new perspective for attention map editing

Weaknesses:
1. Limited handling of non-convex shapes, which may lead to object size reduction or fragmentation
2. May produce unexpected changes or unspecified objects in certain cases
3. Evaluation primarily focuses on SD 1.5, requiring more validation for broader applicability to other T2I models

**Questions For Authors:**

1. How robust is the method when handling complex scenes (such as multiple objects overlapping)? Are there coping strategies?
2. What improvements are being considered for handling non-convex shape objects?
3. How could the method be extended to support simultaneous optimization of multiple text regions?
4. Is adaptive setting of force balance parameters feasible? How could they be automatically adjusted based on scene complexity?

**Relation To Broader Scientific Literature:**

This paper relates to several research directions:

1. Text layout generation: Extends traditional layout design methods, shifting from static image layout to dynamic generation of text-suitable backgrounds.
2. Text-to-image generation: Leverages recent advances in diffusion models but adds consideration for text placement.
3. Attention-guided image editing: Adopts training-free methods to manipulate attention maps, avoiding expensive retraining.

The paper skillfully combines these areas to propose a novel text-friendly image generation method.

**Theoretical Claims:**

The theoretical contributions focus on adapting force-directed graph algorithms, borrowing concepts from physics to apply to attention map editing:

1. Repulsive Force: Ensures each element is placed separately
2. Margin Force: Prevents vertices from being expelled from visual boundaries
3. Warping Force: Maintains object visibility within the canvas through affine transformations

These theoretical claims are clearly understood, logically sound, and well-formulated in the paper.

---

> ### Author Rebuttal · Authors · 2025-03-30
>
> Thank you for your valuable review. We address your concerns with additional experiments and analysis below.
>
> ### [Q1,Q3]Multi-Object and Complex Scene Handling
>
> Our approach shows strong performance in complex scenes with multiple objects. We additionally evaluated on the Desigen dataset, which includes multiple text boxes representing a challenging multi-region scenario:
>
> In multi-object cases, our method creates complementary forces guiding content away from specified regions. This approach extends to complex layouts as shown in our Desigen benchmark results:
>
> | Dataset | Metrics | Desigen training-free | Desigen-Trained | Ours (SD 1.5) |
> |---------|---------|----------------------|-----------------|---------------|
> | Desigen | Saliency IOU ↓ | 40.8 | 41.7 | **32.0** |
> | | TV Loss ↓ | 19.4 | 18.1 | **9.7** |
> | | VTCM ↑ | 2.4 | 2.3 | **3.7** |
>
> In multi-object cases, our method creates complementary forces guiding content away from specified regions.
>
> [W1,Q2] The reviewer may misunderstood our approach. Our warping force and other forces specifically address object size reduction and fragmentation caused by attention reduction, and **this fragmentation equally occurs in both convex and non-convex shapes**. We mentioned non-convex shapes in limitations only to consider more fine-grained region requirements from users, such as spiral-shaped regions.
> For non-convex shapes, our implementation approximates them using their convex hull. While a limitation, user studies show most text and icon placement scenarios involve convex regions, making this practical for common use cases.
>
> [Q4] We're exploring image reward metrics for optimal parameters across different scene complexities, including MLLMs as judges to automatically adjust parameters.
>
> ### [W3] Applicability to Other T2I Models
>
> Our method is extensible to other text-to-image models. We have successfully applied it to multiple diffusion models (see line330 table 2 in the paper).
>
> The implementation is straightforward. After defining the attention processor class, integration requires only a few lines:
> ```python
> for name in pipe.unet.attn_processors.keys():
>     if "attn2" in name:
>         attn_procs[name] = my_attn_processor(guidance_func, region, len(tokens), place_in_unet=name)
>     else:
>         attn_procs[name] = pipe.unet.attn_processors[name]
> pipe.unet.set_attn_processor(attn_procs)
> ```
>
> When combined with the finetuned diffuison model, we see improvements:
>
> | Dataset | Metrics | Desigen-Trained+Attention reduction | Desigen-Trained+Ours | Improvement |
> |---------|---------|-------------------------------------|----------------------|-------------|
> | Desigen benchmark | Saliency IOU ↓ | 41.66 | 38.48 | 7.60% |
> | | TV Loss ↓ | 18.06 | 12.19 | 32.50% |
> | | VTCM ↑ | 2.30 | 2.85 | 23.90% |
> | | CLIP Score ↑ | 28.99 | 26.41 | -8.90% |
>
> ### [Com 3] Computational Complexity and Runtime Analysis
>
> We will add runtime analysis to the appendix. Our method maintains computational complexity comparable to standard diffusion models with minimal overhead from force computations. Our implementation is efficient, requiring less than 15GB VRAM (compatible with consumer GPUs like RTX 3090) and averaging approximately 30 seconds per image generation, as it requires only a single inference pass rather than multiple candidates.
>
> ### [Com 1] Algorithm Pseudocode
>
> We will add simplified pseudocode to method section in the revised paper to clarify the implementation process. Here is the algorithm that summarizes our approach:
>
> ```
> Algorithm: Text-Friendly Background Adaptation
> Input: Text prompt P, Target region R for text placement
> Output: Text-friendly image I with clear space in region R
>
> 1. Initialize diffusion model with prompt P
> 2. For each timestep t:
>    a. Extract cross-attention maps A_k^l for tokens k, layers l
>    b. For each token k, layer l:
>       i. Detect conflicts D(k, R, A_k^l)
>       ii. For conflicting tokens:
>          - Compute attention centroid
>          - Calculate repulsive force F_rep and margin force F_m
>          - Compute total displacement Δpos = F_rep + F_m
>          - Apply warping through affine transformation
>       iii. Apply constraint: A_k,new^l = A_k^l ⊙ (1-R)
>    c. Apply modified maps for guidance
> 3. Return text-friendly image I
> ```
>
> ### Application Scenarios
> We have already supplemented our evaluation with the Desigen dataset(see table 2), which represents poster and advertisement design scenarios. Mobile interface design represents an interesting future direction that we plan to explore.
>
> We appreciate your consideration and helpful suggestions.

---

### Official Review · Reviewer_kAwC · 2025-03-19

**Overall Recommendation:** 3

**Summary:**

This paper aims to solve the problem of generating a background image conditioning on intended size and position of overlaying texts. The proposed approach, TextCenGen, is to generate a regular image and a background image at the same time, and use the text intended region and attention map from the regular image to guide the generation of background image such that it leaves space for texts.

2700 text prompts were used as background captions to generate images. The source of text prompts include ChatGPT generated, Prompt2Prompt, and DiffusionDB. Baselines are other off-the-shelf models in their training-free versions. Qualitatively, the TextCenGen leaves a cleaner space in the background image compared to selected baselines; quantitatively, TextCenGen scores better numbers for lower saliency overlap and higher semantic fidelity.

## update after rebuttal
The provided additional results comparing with baselines in their trained versions and retrieval based method are highly appreciated. This answers my question, and it shows that the proposed method indeed provide additional improvements over existing work. As such, I update my recommendation to 3 weak accept.

**Claims And Evidence:**

One major claim is that TextCenGen outperforms existing methods measured by automatic metrics (saliency IoU, CLIP score, Total Variation Loss, etc.) and user studies. The supporting evidence was 27000 images generated from 2700 selected text prompts, comparing with generic off-the-shelf models, Dalle-3, SD1.5, AnyText, or training-free version of another background generation work, Desigen.

However, given that background generation for texts has its target use cases, like posters and ads, it would be more representative to evaluate on those datasets [1] [2].

Also, even if the proposed method is training free, it would be informative to compare with existing method as is; in particular, to compare with Desigen's trained version.

[1] Weng, Haohan, et al. "Desigen: A pipeline for controllable design template generation." Proceedings of the IEEE/CVF Conference on Computer Vision and Pattern Recognition. 2024.
[2] Zhou, Min, et al. "Composition-aware graphic layout GAN for visual-textual presentation designs." IJCAI, 2022

**Essential References Not Discussed:**

I appreciate the authors including discussions to retrieval based layout planning methods [3], and not just generation based only. It would be interesting to compare with retrieval based methods.

[3] Jin, C., Xu, H., Song, R., and Lu, Z. Text2poster: Laying out stylized texts on retrieved images. In ICASSP, pp. 4823–4827. IEEE, 2022.

**Experimental Designs Or Analyses:**

The dataset to run experiments on could be more relevant to designs. The baseline to compare with could be more informative if they were not limited to their training-free version.

**Methods And Evaluation Criteria:**

Instead of evaluating on generic text prompts, I think it would be more relevant to evaluate on captions from images in layout design datasets [1] [2].

**Other Comments Or Suggestions:**

NA

**Other Strengths And Weaknesses:**

NA

**Questions For Authors:**

NA

**Relation To Broader Scientific Literature:**

If the work conditions not just the text position and size to generate a background, but further conditions on the text font style, color, and its semantic meaning or context, the impact would be broader.

**Theoretical Claims:**

There are no theoretical claims in this paper.

---

> ### Author Rebuttal · Authors · 2025-03-30
>
> Thank you for your valuable review.
>
> ### Evaluation on Poster/Ads Datasets
>
> We selected the Desigen dataset as a layout design dataset for evaluation. We successfully downloaded 53,577 usable images, with 52,806 used for training the Desigen version. The remaining 771 images from the validation set and their corresponding static text masks were used to construct our new Desigen benchmark.
>
> Our core contribution is text-friendly background generation for arbitrary user-specified regions without training constraints. **This approach extends beyond posters and ads to applications like mobile wallpapers where users can specify any position for text, rather than being limited by the distributions in datasets [1,2].**
>
> ### Comparison with Trained Versions of Baselines
>
> Our method focuses on a training-free approach due to the diverse range of user prompts and space region positions. Training often compromises generalization across subtasks.
>
> Table 1 shows Desigen-Trained + Attention Reduction improves over its training-free version on certain metrics. When combining our method with the trained Desigen model, we observe substantial improvements:
>
> | Dataset | Metrics | Desigen-Training-free + Attention Reduction | Desigen-Trained + Atention Reduction | Ours-Replace (SD 1.5) |
> |-|-|--|-|-|
> | P2P Template | Saliency IOU ↓ | 30.6 | 35.0 | **22.9** |
> | | TV Loss ↓ | 13.7 | 14.8 | **8.8** |
> | | VTCM ↑ | 2.9 | 3.0 | **4.4** |
> | DiffusionDB | Saliency IOU ↓ | 30.3 | 31.1 | **23.6** |
> | | TV Loss ↓ | 12.0 | 14.6 | **8.4** |
> | | VTCM ↑ | 3.2 | 2.8 | **4.4** |
> | Synthetics Prompt | Saliency IOU ↓ | 31.6 | 32.6 | **27.7** |
> | | TV Loss ↓ | 15.6 | 14.5 | **11.4** |
> | | VTCM ↑ | 2.7 | 2.9 | **3.5** |
> | Desigen benchmark | Saliency IOU ↓ | 40.8 | 41.7 | **32.0** |
> | | TV Loss ↓ | 19.4 | 18.1 | **9.7** |
> | | VTCM ↑ | 2.4 | 2.3 | **3.7** |
>
> Table above shows Desigen-Trained improves over its training-free version.
>
> When combining our method with the trained Desigen model, we observe substantial improvements:
>
> | Dataset | Metrics | Desigen-Trained+ Attention reduction | Desigen-Trained+Ours |
> |-|-|-|-|
> | Desigen benchmark | Saliency IOU ↓ | 41.66 | **38.48** |
> | | TV Loss ↓ | 18.06 | **12.19** |
> | | VTCM ↑ | 2.3 | **2.85** |
>
>
> ### Comparison with Retrieval-Based Methods
> We evaluated against Text2Poster [3] across all datasets. Our approach differs from retrieval methods: we support arbitrary user regions and create novel content beyond image databases. Since Text2Poster doesn't support user-specified space regions, we tested on preset positions (top, bottom, left, right, center) and selected the best result based on VTCM for fair comparison.
>
> Our related work section (line 083) explicitly mentions Text2Poster, which inspired us to consider background adaptation in the T2I era. We appreciate the reviewer highlighting this discussion direction, as it has provided us with valuable insights.
>
> | Dataset | Metrics | Text2Poster Best@5 | Text2Poster Avg@5 | Ours-Replace (SD 1.5) |
> |-|-|-|-|-|
> | P2P Template | Saliency IOU ↓ | 31.25 | 36.22 | **22.86** |
> | | TV Loss ↓ | 11.49 | 12.45 | **8.81** |
> | | VTCM ↑ | 2.48 | 2.28 | **4.4** |
> | | Clip Score↑ | 20.8 | 20.8 | **27.96** |
> | DiffusionDB | Saliency IOU ↓ | 24.07 | 34.38 | **23.59** |
> | | TV Loss ↓ | **7.99** | 10.65 | 8.41 |
> | | VTCM ↑ | 2.93 | 2.22 | **4.39** |
> | | Clip Score↑ | 17.57 | 17.57 | **27.2** |
> | Synthetics Prompt | Saliency IOU ↓ | 31.25 | 36.91 | **27.7** |
> | | TV Loss ↓ | 11.49 | 13.28 | **11.37** |
> | | VTCM ↑ | 2.48 | 2.16 | **3.49** |
> | | Clip Score↑ | 20.8 | 20.9 | **28.1** |
>
> Our method outperforms retrieval approaches in most text-friendliness metrics while preserving prompt semantics.
> The emergence of autoregressive models like GPT-4o creates potential for combining Text2Poster's retrieval methodology with attention-guided RAG to achieve both text-friendliness and semantic fidelity.
>
>
> ### Broader Impact
>
> Our work extends beyond text placement to other UI elements including icons (Appendix Figure 1). While approaches like TextDiffuser-2 handle text with specific styles through dedicated training, our method achieves user instruction following and arbitrary space region support through attention control. This approach avoids reliance on "sign" or similar elements in prompts (line 088-089), opening a pathway to more flexible controllable generation that preserves both user intent and element compatibility.
>
>
> We appreciate your reconsideration of our paper.
>
> [1] Zheng G, Zhou X, Li X, et al. LayoutDiffusion: Controllable diffusion model for layout-to-image generation. CVPR 2023
>
> [2] Zhou, Min, et al. "Composition-aware graphic layout GAN for visual-textual presentation designs." IJCAI, 2022
>
> [3] Jin, C., et al. Text2Poster: Laying out stylized texts on retrieved images. ICASSP 2022 [PS: Pioneering work]

---

### Decision · Program_Chairs · 2025-05-01

**Decision:**

Accept (poster)

**Comment:**

The reviewers acknowledge the thorough evaluation presented in the paper and the promising performance of the proposed method, supported by subjective, objective, and qualitative results. While there were concerns about the lack of comparisons with highly relevant baselines, evaluation settings that may not fully align with practical applications, and limited evaluation on more recent models, these issues were partially addressed in the rebuttal. Despite these concerns, the reviewers overall gave positive final ratings. The Area Chair agrees with the reviewers and finds the paper to be practical and empirically promising. The paper is considered to be above the acceptance threshold, provided the authors appropriately incorporate the content and promised changes outlined in the rebuttal into the final version.